# A High-Resolution Mass Spectrometer for the Experimental Study of the Gas Composition in Planetary Environments: First Laboratory Results

Illia Zymak [1,2,*], Ján Žabka [3], Miroslav Polášek [3], Arnaud Sanderink [1,4], Jean-Pierre Lebreton [1], Bertrand Gaubicher [1], Barnabé Cherville [1], Anna Zymaková [2] and Christelle Briois [1]

1    Laboratoire de Physique et Chimie de l'Environnement et de l'Espace (LPC2E), UMR7328 CNRS/Université d'Orléans, 3A, Avenue de la Recherche Scientifique, 45071 Orléans, France
2    The Extreme Light Infrastructure ERIC, ELI Beamlines Facility, Za Radnicí 835, 252 41 Dolní Břežany, Czech Republic
3    J. Heyrovský Institute of Physical Chemistry, Czech Academy of Sciences, Dolejškova 3, 182 23 Prague, Czech Republic
4    Institut für Geologische Wissenschaften, Freie Universität Berlin, Malteserstraße 74-100, D-12249 Berlin, Germany
*    Correspondence: illia.zymak@eli-beams.eu

**Abstract:** A new laboratory Orbitrap^TM cell-based mass spectrometer, OLYMPIA (Orbitrap anaLYseur MultiPle IonisAtion), without a C-trap module, has been developed and constructed. The first operation of the Orbitrap^TM cell-based device with the continuous ion source and without the C-trap module is reported. OLYMPIA is being developed and used as a workbench platform to test and develop technologies for the next generation of spaceborne mass spectrometers and as a laboratory instrument to perform high-resolution studies of space-relevant chemical processes. This instrument has been used to measure the quantitative composition of $CO/N_2/C_2H_4$ mixtures of the same nominal mass using an electron ionization ion source. The relative abundance of ions has been measured using a short acquisition time (up to 250 ms) with a precision of better than 10% (for most abundant ions) and a mass resolution of 30,000–50,000 (full width at half maximum) over the mass range of $m/z$ 28–86. The achieved mass accuracy of measurements is better than 20 ppm. This performance level is sufficient to resolve and identify the $CO/N_2/C_2H_4$ components of the mixtures. The dynamic range and relative ion abundance measurements have been evaluated using a reference normal isotopic distribution of krypton gas. The measurement accuracy is about 10% for the 4 most abundant isotopes; 6 isotopes are detectable.

**Keywords:** high-resolution mass spectrometer; Orbitrap; continuous ion source; no C-trap; ion optics



## 1. Introduction

Mass spectrometry is one of the most efficient techniques for in situ experimental studies of the chemical composition of samples of astrochemical relevance. In situ measurements carried out during recent space missions to Saturn and its moons (Cassini-Huygens [1–4]) and the 67P/Churyumov–Gerasimenko comet (Rosetta [5–7]) have provided unique data on the composition of the environment and surface of the Solar System's bodies. These missions unveiled the complexity of the chemical composition of both the atmosphere of Saturn's moon Titan [3,4,8–10] and of comets, which are considered to be relatively primitive small space bodies.

One of the most advanced space mass spectrometers flown onboard space missions, the Rosetta Double Focusing Mass Spectrometer (DFMS) of the Rosetta Orbiter Spectrometer for Ion and Neutral Analysis (ROSINA) module, provided a mass resolution of $m/\Delta m \approx 3000$ at 1% of the peak height (corresponding to $\approx$9000 at full width at half maximum (FWHM)) with a 1–150 u mass range [11–13], which was sufficient to resolve $N_2$ from

CO [14]. Detection of prebiotic organic molecules, such as glycine, was also performed, but deconvolution tools and careful examination of the fragmentation pattern were necessary as the resolution was not sufficient for direct and unambiguous identification of complex organic molecules (Altwegg et al., 2016). For this purpose, when the mass resolution of the measured mass spectra is not sufficient, supplementary analytical deconvolution algorithms [15] and complex chemical models [16,17] are required for a comprehensive description of the chemical composition of extraterrestrial environments.

The chemical diversity of the extraterrestrial medium and the limited mass resolution of current state-of-the-art space instruments call for a new generation of spaceborne mass spectrometers and the development of ground-based analytical instruments for additional comparative laboratory studies of space-relevant samples. High-resolution mass spectrometry allows the identification of complex organic compounds without a need for complementary separation techniques, e.g., chromatography [18], spectroscopy [19], or collision-induced dissociation [20]. In recent years, significant technological and scientific advances have been made in the field of high-resolution space mass spectrometry. Several flight instruments and prototypes of high-resolution mass spectrometers are currently under development and are at different space technology readiness levels (TRL). The MAss SPectrometer for Planetary EXploration (MASPEX) (Brockwell et al., 2016) on board NASA's future Europa Clipper mission is characterized by sufficient mass resolution to identify simple hydrocarbons, i.e., $m/\Delta m \approx 7000$ at $m/z$ 2–32 and around 24,000 at $m/z$ 16–114, using the time-of-flight (ToF) technique. Two high-resolution laser ablation space mass spectrometers, the Characterization of Ocean Residues and Life Signatures (CORALS) instrument and the Characterization of Regolith and Trace Economic Resources (CRATER), are being developed as part of an international collaboration between NASA and CNRS laboratories with the support of CNES [21]. They both include the *CosmOrbitrap*, a space-flight prototype of a high-resolution mass analyzer developed by a consortium of six laboratories [22], based on the commercial Orbitrap$^{TM}$ cell (Thermo Fisher Scientific, Bremen, Germany). The *CosmOrbitrap* has a mass resolution that varies from 474,000 to 90,000 for a 9 to 208 $m/z$ range and an acquisition time of 832 ms [22].

In this work, a novel compact high-resolution mass spectrometry (HRMS) laboratory instrument based on the Orbitrap technology, named OLYMPIA (Orbitrap anaLYseur MultiPle IonisAtion) is presented. The instrument is compatible with different types of ion sources that have recently been built in our laboratory. This setup is being used as a proof of concept for laboratory analytical applications and as a workbench platform. In this work, a configuration that uses a continuous electron ionization (EI) ion source for the high-resolution gas analysis of compounds relevant to planetary environments has been tested. Certain decreases in the mass resolution compared to the ultimate values of the best commercially available Orbitrap mass analyzers are expected due to the higher kinetic energy distribution spread of ions and probable diffusion of the gas from the source inside the trap volume for the proposed configuration of the device. Experimental characterization of the mass analyzer is required to facilitate future use of the proposed configuration for laboratory and space applications.

## 2. Materials and Methods

OLYMPIA (Figure 1) is a HRMS instrument based on the Orbitrap technology for mass analysis proposed and developed by A. Makarov [23]. Its configuration is intended to support the development of future compact and low-resource space instruments without C-traps. The C-trap is a proprietary device used in commercial Orbitrap mass spectrometers for the concentration and increasing monochromaticity of the kinetic energy of ions by collision with the neutral buffer gas prior to injection into an electrostatic trap. It allows achieving ultra-high mass resolution ($m/\Delta m > 10^6$ [24]), but it increases the complexity, mass, and power consumption of the instrument and requires an additional gas reservoir. In contrast, the low-resource design without a C-trap makes OLYMPIA lighter, more transportable, and easier to maintain and operate. It is designed to provide mass resolution

adequate for the next generation of space mass spectrometers ($m/\Delta m$ around 50,000) and is equipped with a modular structure ionization source.

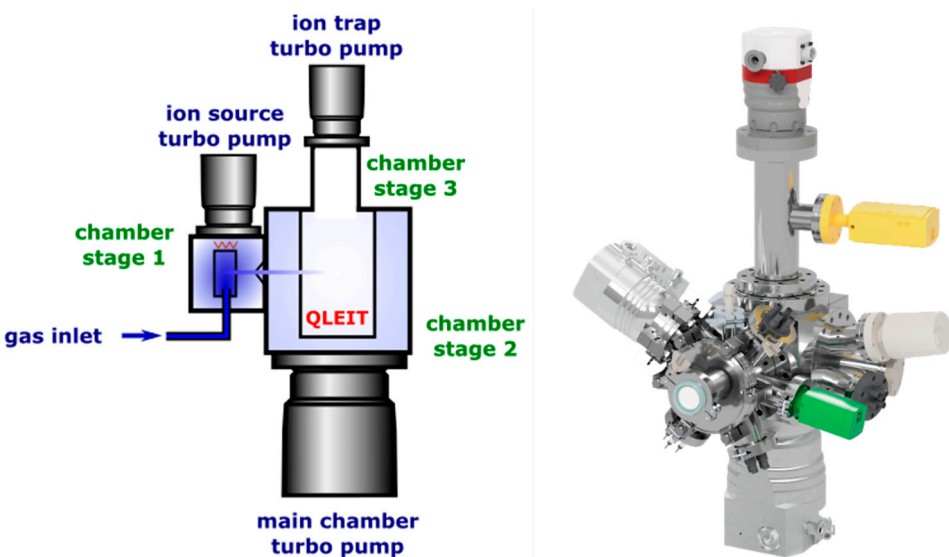

**Figure 1.** Schematic of the OLYMPIA instrument: principal components of the vacuum system (**left** panel); CAD model of the arrangement of the vacuum system (**right** panel) made of Pfeiffer Vacuum, Edwards Vacuum, and Hositrad Vacuum Technologies commercial off-the-shelf components (publicly available models of turbo pumps, vacuum chambers, ion gauges, flanges, and feedthroughs have been used).

*2.1. Ion Source and Ion Optics*

In this work, a continuous EI ion source, built in our laboratory, has been used to produce ions from neutral gas samples. It consists of an ionization chamber with a gas inlet capillary, tungsten filaments, and a pusher electrode that deflects ions towards the ion optics. The variable ionization potential values used for the experiment are within the range of 100–150 V. A gas mixing station was used to prepare samples with a defined mixture ratio. The station is connected to a dry-scroll backing pump and a ceramic technology membrane capacitance pressure gauge (Pfeiffer Vacuum CMR 361, Asslar, Germany). The background pressure of the system, 0.8 mbar, limits the purity of the gas mixture, typically prepared at 1000 mbar. The used gauge ensures equal sensitivity to various gases, and the tolerance of the partial pressure of component gases is rated at 0.2% over the selected pressure measurement range. The gas mixture vessel is first filled with the less abundant gas to improve accuracy. The connecting piping is evacuated to a vacuum level better than is measurable with the gauge after each mixture vessel filling. The commercially available pure gas containers used to prepare the mixture have a purity grade several orders of magnitude higher than the resulting mixture.

Similar to the LAb-CosmOrbitrap (Briois et al., 2016, Selliez et al., 2019), ions are directly injected into the trap from the ion source. For this work, a custom electrostatic ion optics module controlled with high voltage (HV) power supplies (iseg Spezialelektronik GmbH, Germany) was used. The ion optics module is composed of in-line electrodes, as shown in Figure 2. The ion optics is used to transport ions extracted from the ion source, to form the focused bunch of ions, and to correct its pointing if it is needed for the optimal injection conditions. The set of ion lenses is directly connected to the Orbitrap mounting plate. The ion beam is focused into the Orbitrap input orifice and can be detected after passing through the Orbitrap cell using an electron multiplier (DeTech 2700, Detector Technologies, Inc., Palmer, AK, United States) placed behind the Orbitrap, opposite the ion optics. The timing parameters of the injection of an ion packet can be optimized using a pulsed ion optics system controlled by an HV switch module (CGC Instruments, Chemnitz,

Germany). In the current experiment, this option has been used as a transition mode between pulsed and continuous ion source modes.

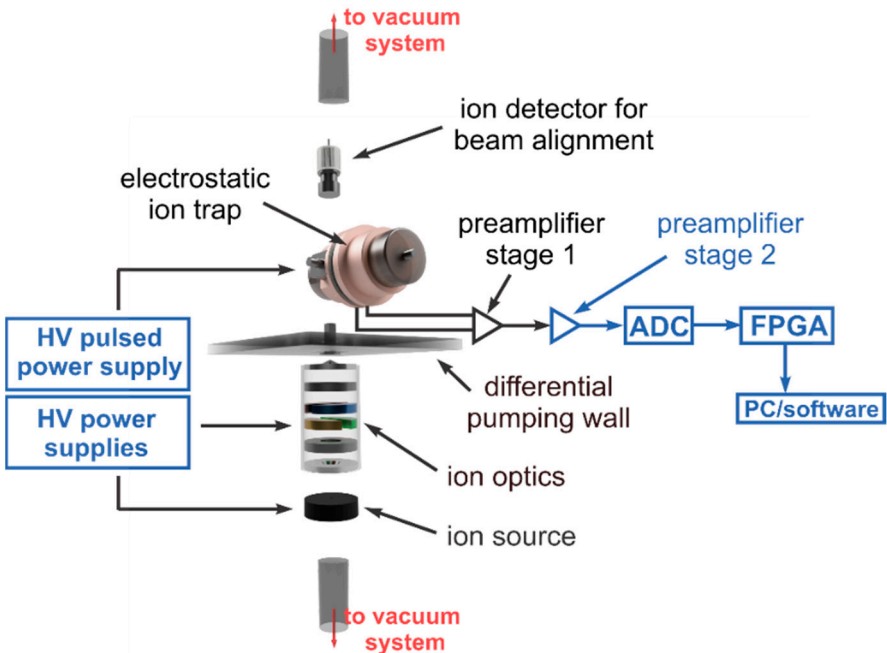

**Figure 2.** Functional illustration of OLYMPIA. Components inside (resp. outside) the vacuum chamber are indicated in grey (resp. in blue).

### 2.2. Mass Analyzer, Data Acquisition and Processing

The core part of the instrument is the D30-type commercial Orbitrap$^{TM}$ cell (Figure 2). Similarly to commercial devices based on the Orbitrap cell, in OLYMPIA, after injection into the trap and electrodynamic squeezing [25], ions are dynamically confined inside the trap by the electrostatic field, spinning around the axis with an oscillating movement along the central electrode axis. The frequency of the cyclic motion of ions is defined by the parameters of the trap and the $m/z$ values of the ions [23].

The axial oscillations of the trapped ions induce the current between the receiving electrodes of the Orbitrap cell. This current is detected by the preamplifier. The signal is then converted by an analog-to-digital converter (ADC) and further processed using Fourier transformation algorithms. Finally, the mass-to-charge ratio of ions is evaluated by the frequency of their oscillation [23].

A new customized high-gain differential preamplifier has been developed and constructed by JanasCard Company (Prague, Czech Republic) for the OLYMPIA setup. This preamplifier has been tested to be functional under vacuum conditions. The preamplifier output voltage is digitized by a 16-bit, 20 MHz, Picoscope$^{®}$ ADC unit (Pico Technology, Cambridge, UK).

The time domain data are pre-processed with doubling signal duration by zero-padding to increase the discrete resolution and, thus, the frequency accuracy. Then, a Hann function-shaped window apodization to minimize spectral leakage [26] is applied. The main data processing includes the application of fast Fourier transformation (FFT) using an FFTW implementation of the Cooley–Turkey algorithm [27] to convert the measured time domain signal into a frequency spectrum. The frequency spectrum is then converted to a mass spectrum using the theoretical model of Orbitrap-based mass analyzers [25,28]:

$$f = \frac{1}{2\pi}\sqrt{\frac{k}{m/q}} \qquad (1)$$

where $f$ is the frequency of ion axial motion, $m$ is the mass of the ion, $k$ is a calibration factor

that characterizes the curvature of the electrostatic field, and $q$ is the electrical charge of the ion. The calibration factor $k$ is calculated for each dataset using the measured frequency and the known exact mass of the calibration ion.

## 3. Experimental Results

Laboratory measurements have been performed to benchmark the instrument's capabilities to resolve isobaric ions in several gas mixtures. Samples composed of molecular nitrogen ($N_2$), carbon monoxide (CO), and ethylene ($C_2H_4$) have been used to determine the resolving power of the mass analyzer at $m/z$ 28. Quantitative measurement capability has been examined with CO and $N_2$ samples at different mixing ratios. Finally, a pure krypton sample was used to assess the quantification of isotopic abundances.

### 3.1. $N_2/CO/C_2H_4$ Gas Mixture

The mass spectra of CO, $N_2$, and $C_2H_4$ gases of the same nominal molecular mass ($m/z$ 28), mixed in equal proportions, have been measured (Figure 3). A high resolution ($m/\Delta m > 3000$ at FWHM) is required to resolve $CO^+$ $m/z$ 27.9949, $N_2^+$ $m/z$ 28.0061, and $C_2H_4$ $m/z$ 28.0313 ions in the spectrum and to measure their respective abundances.

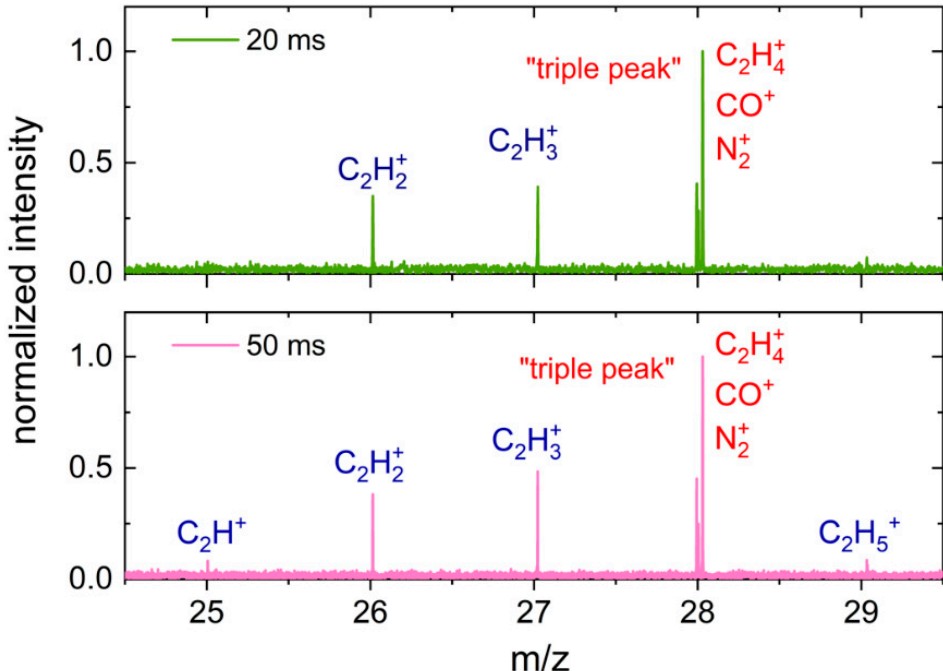

**Figure 3.** The EI mass spectrum of $N_2/CO/C_2H_4$ gas mixture with a 1:1:1 ratio. Ionization chamber pressure is estimated at 1 mbar. Accuracy of the measured peak mass values is 5–20 ppm compared to NIST atomic weights and isotopic composition data. Spectra are evaluated from the same time domain data measurement by adjusting its duration at the processing phase to remain fixed conditions of the experiment.

The different sampling times from 20 to 500 ms have been used to measure mass spectra for the estimation of the optimal value for the current configuration of OLYMPIA. The maximal reasonable signal duration is about 250 ms due to the relatively high pressure inside the ion trap chamber, approximately $10^{-8}$ mbar. Longer acquisition reduces the total amplitude of the effective signal and, thus, the signal-to-noise ratio and dynamic range.

The partial mass spectra of the $CO/N_2/C_2H_4$ mixture, are presented in Figure 3. The mass accuracy of the measured peaks varies from 5 to 20 ppm compared to NIST atomic weights and isotopic composition data. A zoom of the region around $m/z$ 28 is shown in Figure 4. Three different peaks corresponding to $CO^+$, $N_2^+$, and $C_2H_4^+$ have been successfully resolved. A mass resolution of about 40,000 at $m/z$ 28 has been achieved with

a sampling time of less than 250 ms (Figure 4). In Figure 4, a combination of increasing resolution and variation in abundance accuracy is observed for a longer acquisition time.

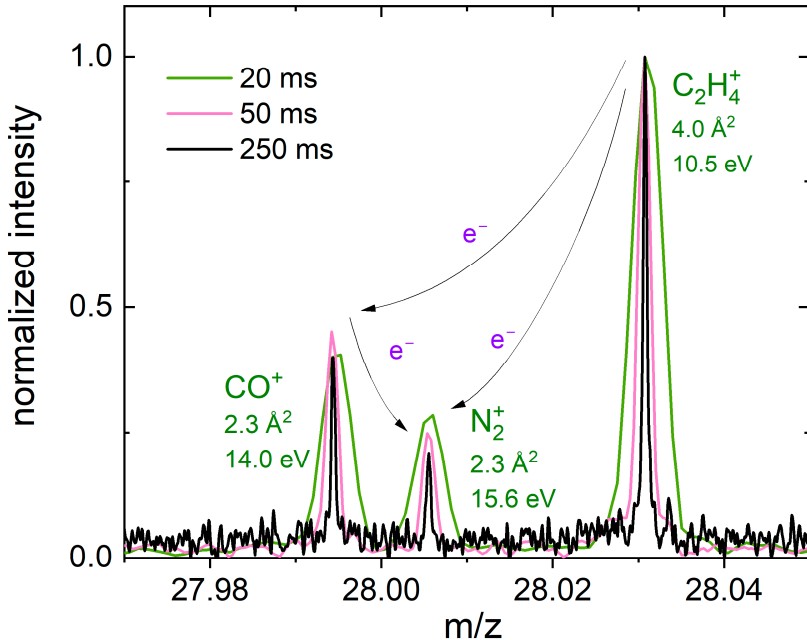

**Figure 4.** Mass spectra of the $N_2/CO/C_2H_4$ mixture measured with different FFT duration times of the same time domain measurement ($m/\Delta m \approx 6000$ at 20 ms, $m/\Delta m \approx 16,000$ at 50 ms, $m/\Delta m \approx 47,000$ at 250 ms). The ionization cross-section of neutrals and the ionization energy are given for the three species. Direction of the charge transfer reaction is indicated using the arrows.

The relative abundance of ions, which can be determined from the measured spectrum as peak amplitudes, does not only depend on the initial partial concentration of gases in the mixture. The ionization cross-section (1) and further chemical processes inside the ion source, such as charge transfer (2) and fragmentation (3), have a strong impact on the quantitative composition of the ion bunch injected into the trap.

(1)    According to the binary encounter Bethe (BEB) model [29], the electron ionization cross-section, and thus the ionization probability of the $C_2H_4$ molecules (4.0 Å$^2$), is almost twice that of the CO and $N_2$ molecules (2.3 Å$^2$) (NIST Standard Reference Database 107 data).

(2)    After the ionization at low vacuum conditions of the ion source (the pressure is about 1 mbar inside the ionization chamber), charge transfer reactions occur. Cations with lower ionization energies of precursor neutrals (CO—14.0 eV, $C_2H_4$—10.5 eV, $N_2$—15.58 eV) [30–32] are produced more effectively.

(3)    Ions $C_2H_4^+$, their fragments $C_2H^+$, $C_2H_2^+$, or $C_2H_3^+$, and secondary reaction products $C_2H_5^+$ can be produced from $C_2H_4$ molecules (Figure 4). Fragmentation of neutrals and reactions of $C_2H_4^+$ ions reduce the relative $C_2H_4^+$ amplitude.

The combination of these three factors results in a higher abundance of $C_2H_4^+$ and a lower abundance of CO$^+$ and $N_2^+$ ions at equal concentrations of neutrals (Figure 5). The exact current of ions extracted from the source and its ratio is not constant and depend on the parameters of the setup, such as the pressure in the source and the accelerating potential of ionizing electrons. Therefore, calibration measurements at relevant conditions are required for quantitative in situ characterization of the sampled gas mixtures.

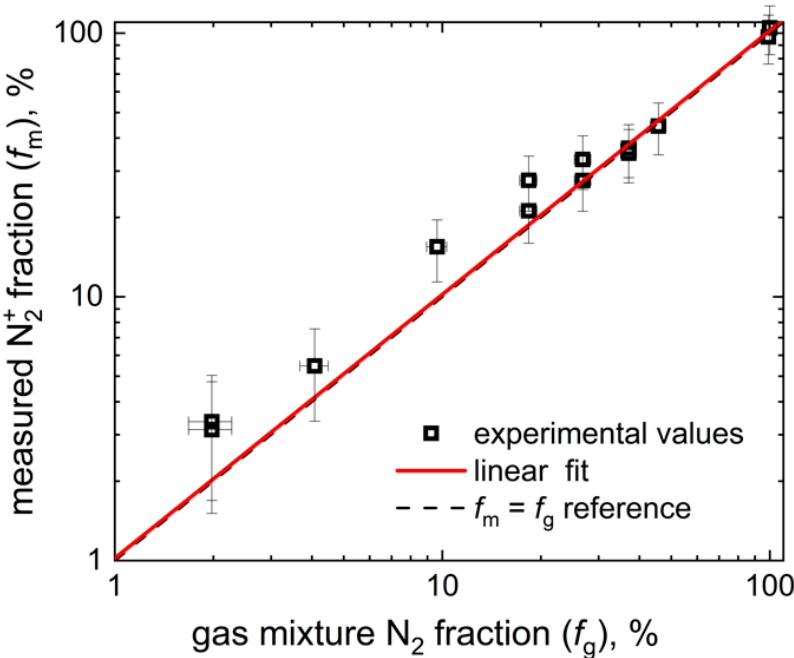

**Figure 5.** Measured fraction of $N_2^+$ ions as a function of nitrogen gas ratio in the known mixture. The fit function is calculated for the linear scale data using the least squares method.

### 3.2. CO/$N_2$ Gas Mixture

CO and $N_2$ gases of similar ionization cross-sections mixed in different ratios have been used to examine the quantitative measurement capability of OLYMPIA. These molecules have the same ionization cross-section and so does the probability of the production of ions by electron–neutral interactions, according to the BEB model. Nevertheless, they have different ionization energies, which affect further chemical reactions and the ratio of ions extracted from the source.

Several mixtures with CO fractions ranging from 2% up to 98% in the $N_2$ gas have been used (Figure 5). The relative intensity of $CO^+$ in $N_2^+$ ions was measured with a mass resolution sufficient to identify ions of the same molecular mass ($m/\Delta m \approx 5000$). A trace amount of carbon monoxide gas was found to be detectable in mixtures with a relative abundance of less than 2%.

Measurement of the CO/$N_2$ mixture showed an estimation of species abundance over two orders of magnitude of the CO concentration. At low $N_2$ concentrations, the measured $N_2^+$ to $CO^+$ ion ratio is slightly higher (but within the range of the statistical error) than the neutral gas ratio in the prepared mixture. The higher number of detected $CO^+$ ions can also be explained by the higher resulting efficiency of ion production due to the charge transfer processes, compared to Figure 4. The measurable amounts of $C^+$, $O^+$, and $N^+$ fragment ions were not observed.

### 3.3. Krypton Gas

The sensitivity of the instrument has been verified using a polyisotopic monoatomic gas to improve the amplitude measurement accuracy. The use of one monoatomic gas eliminates the effects of further chemical reactions of formed ions and fragmentation of neutrals. Meanwhile, the isotopic composition of krypton allows to estimate the accuracy of the quantitative characterization of the sample over a broad range of ratios. Krypton gas with 99.998% purity and natural isotopic abundance was selected as a calibrant as it has a well-documented isotopic ratio. It has also been detected in the coma of the 67P/Churyumov–Gerasimenko comet [14], so the study of the possibility of accurate measurement of its isotopic composition is of planetary science interest. A krypton gas

container could also be proposed to be used as an onboard calibrant for future space missions that would use an Orbitrap-based spectrometer.

Spectra were measured with a 40 ms acquisition time (Figure 6). All six main isotopes of Krypton, even $^{78}Kr^+$, which is only 0.355% abundant, were successfully detected. The statistical error of the $^{82}Kr$, $^{83}Kr$, $^{84}Kr$, and $^{86}Kr$ peak amplitude measurements was 9–11%.

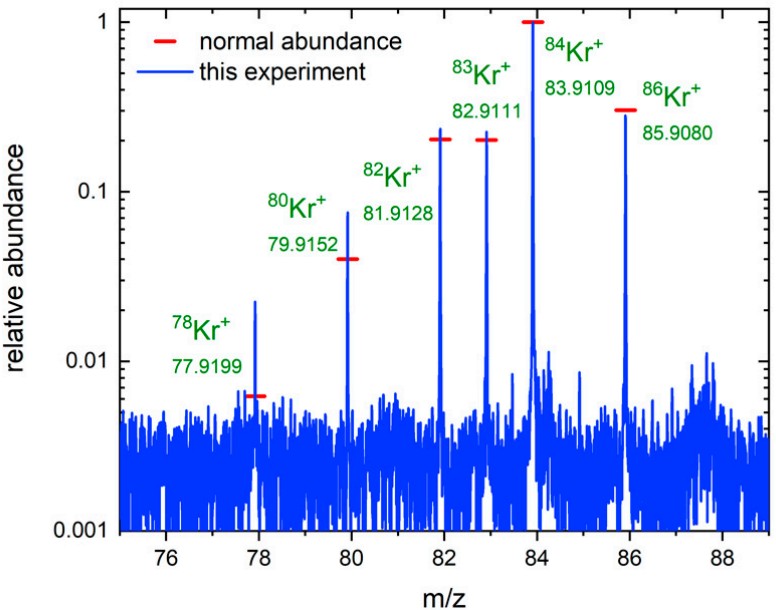

**Figure 6.** Mass spectra of the krypton gas calibrated and normalized to $^{84}Kr^+$ ion compared to the NIST isotopic composition database (red marks). Experimental $m/z$ values are indicated near labels of isotopes by green numbers (better than 20 ppm mass accuracy).

A higher-than-expected amplitude has been observed in the OLYMPIA data for the least abundant ions of the krypton gas $^{78}Kr^+$ and $^{80}Kr^+$ (Figure 5) and for the $CO/N_2$ mixture (Figure 6). The measured difference from the normal isotopic distribution for less abundant ions is statistically significant. It indicates an accuracy limit for the measurement of trace amounts of components in the studied samples for the current configuration of the mass analyzer.

## 4. Outlook for Limited-Resources Space Missions

OLYMPIA is one of the laboratory concepts for future compact Orbitrap-based HRMS instruments for space missions. It specifically targets low-resource missions, including CubeSat-class ones, e.g., the currently developed Space Laboratory for Advanced Variable Instruments and Applications (SLAVIA) space mission dimensioned to 12U size and less than 20 kg mass format. The constraints imposed by the space environment (miniaturization, pressure, low power consumption, limited data transfer, especially for outer solar system bodies) have been considered during the development of this prototype.

For the OLYMPIA setup, the choice was made to not include a C-trap module present in commercial Orbitrap in order to reduce the overall complexity of the space instrument. The feasibility of this configuration had already been proven for pulsed ion sources using the LAb-CosmOrbitrap [22]. In this work, it has been demonstrated that the proposed configuration is compatible with a continuous EI source while ensuring the sufficient performance expected for HRMS.

A mass resolution of 50,000 is desirable to obtain a new level of scientific output and state-of-the-art knowledge for the extraterrestrial medium composition. Selection of the optimal sampling duration for specific tasks of the mission, and target mass resolution, is an important phase of the instrument design. Effective sampling duration is defined by (a) the time stability of power supply designed for space applications (e.g., compact HV

source for CubeSats), (b) the pressure in the vicinity of the spacecraft (e.g., in low Earth orbit), and (c) the computing resources required to apply the FFT on board.

Finally, for most space missions, and in particular for low-resource CubeSats, data transfer is highly limited by the transmission capability of the spacecraft. In order to meet the challenges resulting from the proposed space mission application, a trade-off on the optimal signal duration is required to ensure sufficient mass resolution, ion abundance accuracy, and data handling. While the short signal duration decreases the mass resolution, it improves the signal-to-noise ratio and limits computations and data transfer requirements (Figure 4).

### 5. Conclusions

A new Orbitrap-based HRMS laboratory instrument has been built and characterized. The first operation of the Orbitrap-cell mass analyzer without a C-trap module with the continuous EI ion source has been demonstrated. Its performance has been evaluated by the analysis of gas mixtures composed of chemical species of the same nominal molecular mass (CO, $N_2$, and $C_2H_4$). The achieved mass resolution for $^{14}N_2^+$ is more than 40,000 for a 250 ms FFT duration. The experimental resolution value, limited by the pressure inside the Orbitrap cell, is close to the value of about 50,000 (for at least 2–3 points per peak) defined by the FFT theory for this acquisition time and allows to resolve all three chemical species without any additional analytical techniques. With a mixture of CO and $N_2$ at different mixing ratios and a pure krypton gas sample, the ability of OLYMPIA to measure ion abundances over two orders of magnitude has been assessed with an accuracy of 5–50% from the most abundant to the least abundant fractions. Despite a higher-than-expected measured intensity for less abundant ions (oversensitivity), the error range is less than 10% for the most abundant ions. The dynamic range of the instrument allows the detection of chemical species with a volume ratio of less than 1%. Indeed, the krypton isotope ($^{78}Kr$) with an abundance of less than 0.4% is clearly detectable in the spectrum with a signal-to-noise ratio of better than 250 for the major isotope.

The performed experiment demonstrates the technical feasibility of space applications of the OLYMPIA setup using a configuration of Orbitrap-based mass spectrometers without a C-trap module (see Outlook for limited-resource space missions). It can be used with continuous or pulsed ion sources, even for applications with high mass resolution requirements. This instrument configuration will reduce mechanical, high-voltage control, and timing module complexity while providing reasonable performances and a mass resolution exceeding that of previous space missions by at least one order of magnitude. The experiment also facilitates the analysis of ions abundant in atmospheres; however, the sensitivity and signal-to-noise ratio of the system should be improved.

With the current mass analyzer configuration, OLYMPIA proves to be an efficient tool, with certain trade-offs related to simplified system design, to study complex gas mixtures relevant to the atmospheres of the Solar System bodies (e.g., Titan), including the Earth, and can be used as a research and development workbench for future space missions.

**Author Contributions:** I.Z.—design of the experiment, optimization and construction of the experimental setup, data analysis, editing of the manuscript; J.Ž.—design and construction of the experimental setup, data analysis, editing of the manuscript; M.P.—experiment design, data analysis, editing of the manuscript; A.S.—laboratory work, editing of the manuscript; J.-P.L.—data analysis, editing of the manuscript; B.G.—construction of the experimental setup; B.C.—laboratory work; A.Z.—editing of the manuscript; C.B.—group leader, project management, editing of the manuscript. All authors have read and agreed to the published version of the manuscript.

**Funding:** We acknowledge support from LE STUDIUM Smart Loire Valley Programme/European Union Horizon 2020 research and innovation program under the Marie Sklodowska Curie grant agreement # 665790 and the CNES Research and Technology program. This work was also supported by the Czech Science Foundation (grant No. 21-11931J).

**Data Availability Statement:** The data presented in this study are available on request from the corresponding author. The data are not publicly available due to use of sensitive information provided by Thermo Fisher Scientific, Bremen, Germany.

**Acknowledgments:** Authors of the paper gratefully acknowledge Alexander Makarov for his kind support, starting from helping with the conceptual design of a new instrument to the final data analysis. This support motivated the research team to try new methods and approaches.

**Conflicts of Interest:** We have no conflict of interest to disclose.

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
