# Peer review of "A High-Resolution Mass Spectrometer for the Experimental Study of the Gas Composition in Planetary Environments: First Laboratory Results"

_aerospace, doi:10.3390/aerospace10060522_

Round 1
Reviewer 1 Report
Designing and using (ultrahigh) resolution mass spectrometers in space is absolutely essential to obtain correct and reliable information on atmospheric or coma components, including (small) organic materials. Although low resolution mass spectrometers provided good basic data, e.g., in the atmosphere of Titan, some assignments were not correct due to lack of high resolution and also because of the existence of protonated, nitrogen containing compounds (that were assigned as "Cn carbons").
I welcome the work summarized nicely in the submitted manuscript. However, I have a few comments for the authors' consideration:
i) The accurate masses of the CO, N2 and C2H4 ions (generated by electron ionization) are not correct, the electron mass is missing in all cases. Correctly, these ion masses are: 27.994366 (CO), 28.005599 (N2) and 28.03752 (C2H4). (Note that the electon's mass is ca. 0.55 mdalton.)
ii) It is not important but it would be nice to show the order of CO<N2<C2H4 in the text according to their increasing m/z.
iii) What was the EI ionization (electron) energy? 70eV or lower, or variable? It may have some importance in "space" studies, please make a comment on this.
iv) What was the pulsing time from the continuous EI source? Synchronized with the time domain time (e.g., 20, 50, 250 ms)? It would be desirable to provide more details about "pulsing" (e.g., on the "ion optics" shown in Figure 2).
v) The use of EI source is inevitable when one wants to study neutral components in atmospheres. But there are a lot of ions in planetary or coma atmospheres, just as detected by Cassini's INMS (Ion/Neutral MS). It would be important to give a perspective, how the current design could be used to detect already existing ions in atmospheres. Can you collect or "trap" them (not in a C-trap but like in an ion funnel)? I understand that this was not the focus of your work but detection ions (without ionization) is an important part of planetary atmospheric studies.
vi) Use "Figure" consistently (i.e., start with a capital F).
Author Response
Reply to referee 1
Dear referee, thank you for reading our paper! We appreciate proposing your important suggestions and comments. All of your suggestions are implemented and-or commented on.
Details on implemented corrections are listed below.
Designing and using (ultrahigh) resolution mass spectrometers in space is absolutely essential to obtain correct and reliable information on atmospheric or coma components, including (small) organic materials. Although low resolution mass spectrometers provided good basic data, e.g., in the atmosphere of Titan, some assignments were not correct due to lack of high resolution and also because of the existence of protonated, nitrogen containing compounds (that were assigned as "Cn carbons").
I welcome the work summarized nicely in the submitted manuscript. However, I have a few comments for the authors' consideration:
- i) The accurate masses of the CO, N2 and C2H4 ions (generated by electron ionization) are not correct, the electron mass is missing in all cases. Correctly, these ion masses are: 27.994366 (CO), 28.005599 (N2) and 28.03752 (C2H4). (Note that the electon's mass is ca. 0.55 mdalton.)
Thank you for pointing us to this mistake. Indeed, the neutral masses are indicated in the text, however, labeled as ions. Corrected in the text according to the suggestion. The ion, not neutral masses have been used for the data processing.
- ii) It is not important but it would be nice to show the order of CO<N2<C2H4 in the text according to their increasing m/z.
Corrected in the text according to the suggestion.
iii) What was the EI ionization (electron) energy? 70eV or lower, or variable? It may have some importance in "space" studies, please make a comment on this.
The ionization potential values used in the experiment are 100 – 150V. That unusually high value and mbar-range pressure inside the ionization chamber have been used to increase the ion current from the source and to improve SNR of spectra.
- iv) What was the pulsing time from the continuous EI source? Synchronized with the time domain time (e.g., 20, 50, 250 ms)? It would be desirable to provide more details about "pulsing" (e.g., on the "ion optics" shown in Figure 2).
In the current research all presented experimental data has been measured for the continuous EI source and modes optics. It has been powered with DC voltage supplies. The only electrodes to which pulsed potentials have been applied are the internal electrodes of the Orbitrap (switching from the injection to trapping mode), according to the technique of trapping and electromagnetic squeezing of the injected ions inside the trap. The typical time of switching between the injection and trapping potentials is in the range of several hundreds of microseconds.
- v) The use of EI source is inevitable when one wants to study neutral components in atmospheres. But there are a lot of ions in planetary or coma atmospheres, just as detected by Cassini's INMS (Ion/Neutral MS). It would be important to give a perspective, how the current design could be used to detect already existing ions in atmospheres. Can you collect or "trap" them (not in a C-trap but like in an ion funnel)? I understand that this was not the focus of your work but detection ions (without ionization) is an important part of planetary atmospheric studies.
Thank you for addressing an interesting question According to the current performance of the OLYMPIA setup, the efficiency of the collection of atmospheric ions should be such that it provides at least nA ion current. The “very optimistic” volume number density values for positive ions in the atmosphere of the Earth can be estimated as 1e6 cm-3. Thus, the ions would have to be collected from a volume of at least 1e4 cm3 per second to produce the required current. We consider it as theoretically possible, but for a technical reason certain improvement in the sensitivity of the device and efficient ion concentration technique, e.g. DC-carpet is required.
As mentioned in the text according to the suggestion
- vi) Use "Figure" consistently (i.e., start with a capital F).
Corrected in the text according to the suggestion
Reviewer 2 Report
Zymak et al. presents a performance assessment study of a new lightweight orbitrap instrument for (molecular) gas analysis of planetary objects. This instrument is a welcome addition to the family of space orbitrap systems, which thus far have mainly focussed on surface analysis in combination with laser ablation/desorption techniques. There are a number of relevant follow-up measurements that come to mind that can be done with this instrument, such as trace gas analysis, Earth atmosphere measurements, or the measurement of complex organic molecules (or mixtures thereof) in the 50 - 100 u range. Perhaps some of these measurements could still be done for the current publication to add more body to the results section, but the overall body of work in the manuscript is acceptable for publication.
Before recommending this manuscript for publication, I want to ask the authors to address several questions/issues, which primarily relate to the level of detail. In several cases information that I find critical is not found in the manuscript or cannot be found in the cited literature (see point below about the Žabka 2018 reference).
My main comments are listed below, followed by a number of minor or grammar comments.
#############################################################
MAIN COMMENTS.
Section 2.1: I don't find any mention of the electron ionization energy used. Is this the typical 70 eV?
Section 2.1: More information on the gas mixing protocol/technique is required. Because section 3.2 relies on accurate knowledge of the CO:N2 ratio, down to 2% levels, I find it relevant to show that a mixing protocol/technique is used that accurately can achieve such mixing ratios.
Section 2.1: The authors refer to Žabka 2018 for details regarding the ion source and optics. I get the impression that this paper is a chapter in a conference proceedings, but I have been unable to find it - also not via the Le Studium website. Can the authors include a doi or arxiv link, so it is easier for me (and future readers) to locate this publication? I think this is particularly important for the reader interested in the performance of the ion optics and the characteristics of the ion package, which ultimately affect the performance of the orbitrap.
Section 3.1 + Figure 3: By eye, there seem to be variations in the signal intensities of the C2H4 fragment ions at m/z 26 and 27 between the 20 and 50 ms acquisition (27/26 for 50 ms > 20 ms run). I suspect this might have to do with effects like charge transfer, fragmentation, and secondary product formation mentioned later in this section. However, I still find this peculiar if the conditions (e.g., mixing ratios, gas inlet pressure, etc) remained constant between both acquisitions. Can the authors comment on the parameters that affect the ratios of these C2H4 fragments? (or tell me if my by-eye assessment is simply incorrect)
Section 3.2: Several things are unclear in this section, and additional information should be added to clarify the text. 1) For clarity, I suggest the authors explicitly mention that they measure the intensity of the N2+ and CO+ ions. 2) The fragmentation patterns of the two molecules into C+, O+ and N+ also affects the ion signal and ratios. Is this taken into account? How? 3) Do the authors mean mixing ratios or fractions (as mentioned in Fig. 5)? 4) I do not understand why the Zymak et al. 2021 reference - which is a conference abstract - is cited in the results section of this manuscript. In my opinion, this reference should be removed. 5) Details about the linear fit and its use should be added. Is the fit made in log or linear scale?
Section 4, paragraph 1: A number of statements in this paragraph are not well supported by the provided information in the manuscript. While I am willing to believe that the orbitrap fits in a CubeSat, I have doubts about the complete package, including ion source and optics + electronics. There is also no information in the manuscript about the power consumption and amount of data generated per measurement. A table that lists the approximate mass, size, and power consumption of the components of this instrument, plus additional information in the manuscript would be helpful to support this paragraph.
#############################################################
MINOR / GRAMMAR COMMENTS.
Overall grammar check: I've noticed several instances where the article before the noun is missing, singular / plural are misused, commas are missing/overused, or sentences are too long (introduction, paragraph 2 starts with a sentence spanning 6 lines). This is not critical, but I had to read some sentences twice before I understood them. To improve the reading experience, I would suggest performing an additional grammar check of the manuscript (for example with Grammarly, which I also used to check the grammar in this report).
Title: The word order of the title seems a bit off to me. I suggest the following as an alternative: "A high-resolution mass spectrometer for the experimental study of the gas composition in planetary environments"
Abstract, sentence 1: Comma usage in this sentence makes it hard to read - consider splitting it into two sentences.
Abstract, sentence 2: repetition w.r.t. sentence 1.
Abstract, sentence: "This performance level is sufficient to resolve and identify N2/CO/C2H4 components of the mixtures." I think it is worth adding to this sentence that these molecules have the same nominal mass to emphasize that a high-mass-resolution instrument is required to identify these compounds.
Introduction, sentence 2 contains mission names and references in double brackets.
Introduction, sentence "The MAss SPectrometer for Planetary EXploration (MASPEX) (Brockwell et al., 2016) on board NASA’s future Europa Clipper mission ... ": For the reader, it might be helpful to mention the MS technique (ToF) used in this instrument.
Materials, Fig. 2: Can the authors add if all components are shown to the same scale (specifically, the trap, ion optics, detector, and source). If not, can approximate size/scale ratios between the components be included so the reader can get a feeling for the overall size.
Figure 4: The red / green labeling initially confused me a bit. I suggest to align the cross-section and IP text below the relevant molecule and giving it the same green color. The meaning of the electrons and arrows is not clear to me.
Section 3.3, sentence: "so study of the possibility of accurate measurement of its isotopic composition is of astrochemical interest." In my opinion, it is more appropriate to place this under the general interests of the planetary science community. Astrochemical interests primarily cover molecules after all.
Figure 6: I find the mention of "0.4% fraction" for 78Kr when the figure is given in abundances normalized to 84Kr a bit confusing. I think it is better to consistently represent the Kr isotope values as either fractions or abundances, but not a mix of both. This also counts for the manuscript text in 3.3. Furthermore, to better see the 78Kr signal, I would suggest to plotting the figure in log scale or adding a zoom-in window.
Section 5, sentence: "... has been assessed with pretty good accuracy." Pretty good accuracy is a subjective description. I suggest simply mentioning the values at this point.
Author Response
Reply to Referee 2
Dear referee, we highly appreciate you rigorous reading of our manuscript and your valuable work aimed to improve readability and scientific quality of the paper. We find all of your comments and raised up discussions reasonable and important.
We agree that some data can be enhanced with additional experiments, but currently the instrument is moved to another country for water or solid samples experiments.
Please find details listed below.
#############################################################
MAIN COMMENTS.
Section 2.1: I don't find any mention of the electron ionization energy used. Is this the typical 70 eV?.
No, it was 100-150 eV. The manuscript is enhanced according to the suggestion.
Section 2.1: More information on the gas mixing protocol/technique is required. Because section 3.2 relies on accurate knowledge of the CO:N2 ratio, down to 2% levels, I find it relevant to show that a mixing protocol/technique is used that accurately can achieve such mixing ratios.
The manuscript is enhanced according to the suggestion.
Section 2.1: The authors refer to Žabka 2018 for details regarding the ion source and optics. I get the impression that this paper is a chapter in a conference proceedings, but I have been unable to find it - also not via the Le Studium website. Can the authors include a doi or arxiv link, so it is easier for me (and future readers) to locate this publication? I think this is particularly important for the reader interested in the performance of the ion optics and the characteristics of the ion package, which ultimately affect the performance of the orbitrap.
The reference suggested as inaccessible/doubtful has been removed, as we believe that cited paper dose not contribute significantly to the clarity of the paper. It will be more relevant to the pulsed beam injection mode. In the continuous mode, the ion optics mostly affects an alignment of the ion beam, while the timing of the ion injection into the Orbitrap depends on the Orbitrap electrodes timing.
Section 3.1 + Figure 3: By eye, there seem to be variations in the signal intensities of the C2H4 fragment ions at m/z 26 and 27 between the 20 and 50 ms acquisition (27/26 for 50 ms > 20 ms run). I suspect this might have to do with effects like charge transfer, fragmentation, and secondary product formation mentioned later in this section. However, I still find this peculiar if the conditions (e.g., mixing ratios, gas inlet pressure, etc) remained constant between both acquisitions. Can the authors comment on the parameters that affect the ratios of these C2H4 fragments? (or tell me if my by-eye assessment is simply incorrect)
Data given at the figure is actually evaluated from the same data acquisition (time domain measurement) changing (shortening) length of the data for FFT transformation. So, we are actually analysing the same bunch of ions injected into the trap, and observing effects of processes inside the trap. With this we exclude variability of the experimental conditions, but indicate limitations (actually weakness) of our configuration of the Orbitrap.
This issue is now explained clearer in the manuscript.
Section 3.2: Several things are unclear in this section, and additional information should be added to clarify the text. 1) For clarity, I suggest the authors explicitly mention that they measure the intensity of the N2+ and CO+ ions. 2) The fragmentation patterns of the two molecules into C+, O+ and N+ also affects the ion signal and ratios. Is this taken into account? How? 3) Do the authors mean mixing ratios or fractions (as mentioned in Fig. 5)? 4) I do not understand why the Zymak et al. 2021 reference - which is a conference abstract - is cited in the results section of this manuscript. In my opinion, this reference should be removed. 5) Details about the linear fit and its use should be added. Is the fit made in log or linear scale?
- The manuscript is changed according to your suggestion.
- We did not observe detectable number of fragment ions, so we neglect the fragmentation effects, however weacknowledge that certain mass discrimination effect is possible, so number of fragment ions is underestimated.
- There has been meant the fraction. The manuscript is changed according to your suggestion.
- The reference suggested as inaccessible/doubtful has been removed.
- There has been used least square roots algorithm applied to the linear scale data. The manuscript is changed according to your suggestion.
Section 4, paragraph 1: A number of statements in this paragraph are not well supported by the provided information in the manuscript. While I am willing to believe that the orbitrap fits in a CubeSat, I have doubts about the complete package, including ion source and optics + electronics. There is also no information in the manuscript about the power consumption and amount of data generated per measurement. A table that lists the approximate mass, size, and power consumption of the components of this instrument, plus additional information in the manuscript would be helpful to support this paragraph.
The manuscript is enhanced according to your suggestion, to specify more clearly what is meant as a compact Cubesat-class (actually 12U) satellite.
#############################################################
MINOR / GRAMMAR COMMENTS.
Overall grammar check: I've noticed several instances where the article before the noun is missing, singular / plural are misused, commas are missing/overused, or sentences are too long (introduction, paragraph 2 starts with a sentence spanning 6 lines). This is not critical, but I had to read some sentences twice before I understood them. To improve the reading experience, I would suggest performing an additional grammar check of the manuscript (for example with Grammarly, which I also used to check the grammar in this report).
Thank you for the suggestion, manuscript has been checked using the Grammarly application.
Title: The word order of the title seems a bit off to me. I suggest the following as an alternative: "A high-resolution mass spectrometer for the experimental study of the gas composition in planetary environments"
The title has been changed according to the suggestion.
Abstract, sentence 1: Comma usage in this sentence makes it hard to read - consider splitting it into two sentences.
Changed according to the suggestion
Abstract, sentence 2: repetition w.r.t. sentence 1.
Changed according to the suggestion
Abstract, sentence: "This performance level is sufficient to resolve and identify N2/CO/C2H4 components of the mixtures." I think it is worth adding to this sentence that these molecules have the same nominal mass to emphasize that a high-mass-resolution instrument is required to identify these compounds.
Changed according to the suggestion
Introduction, sentence 2 contains mission names and references in double brackets.
Changed according to the suggestion
Introduction, sentence "The MAss SPectrometer for Planetary EXploration (MASPEX) (Brockwell et al., 2016) on board NASA’s future Europa Clipper mission ... ": For the reader, it might be helpful to mention the MS technique (ToF) used in this instrument.
Changed according to the suggestion
Materials, Fig. 2: Can the authors add if all components are shown to the same scale (specifically, the trap, ion optics, detector, and source). If not, can approximate size/scale ratios between the components be included so the reader can get a feeling for the overall size.
Changed according to the suggestion
Figure 4: The red / green labeling initially confused me a bit. I suggest to align the cross-section and IP text below the relevant molecule and giving it the same green color. The meaning of the electrons and arrows is not clear to me.
The labels are changed according to the suggestions. The meaning of the arrows that indicate the direction of the charge transfer reactions is explained in the figure caption.
Section 3.3, sentence: "so study of the possibility of accurate measurement of its isotopic composition is of astrochemical interest." In my opinion, it is more appropriate to place this under the general interests of the planetary science community. Astrochemical interests primarily cover molecules after all.
Changed according to the suggestion
Figure 6: I find the mention of "0.4% fraction" for 78Kr when the figure is given in abundances normalized to 84Kr a bit confusing. I think it is better to consistently represent the Kr isotope values as either fractions or abundances, but not a mix of both. This also counts for the manuscript text in 3.3. Furthermore, to better see the 78Kr signal, I would suggest to plotting the figure in log scale or adding a zoom-in window.
Section 5, sentence: "... has been assessed with pretty good accuracy." Pretty good accuracy is a subjective description. I suggest simply mentioning the values at this point.
Changed according to the suggestion
Round 2
Reviewer 2 Report
I thank the authors for making the adjustments to their paper and I am happy to recommend this manuscript for publication.